# Molecular Tumor Board-Guided Targeted Treatments for Biliary Tract Cancers in a Publicly Funded Healthcare System

**DOI:** 10.3390/curroncol32020080

**Published:** 2025-01-31

**Authors:** Felix E. G. Beaudry, Zhihao Li, Ayelet Borgida, Anudari Zorigtbaatar, Xin Wang, Maggie Hildebrand, Oumaima Hamza, Gun Ho Jang, Roxana Bucur, Anna Dodd, Julie Wilson, Rebecca C. Auer, Samuel Saibil, Erica S. Tsang, Arndt Vogel, Grainne M. O’Kane, Steven Gallinger, Jennifer J. Knox, Faiyaz Notta, Gonzalo Sapisochin, Robert C. Grant

**Affiliations:** 1Ontario Institute for Cancer Research, Toronto, ON M5G 1M1, Canada; fbeaudry@oicr.on.ca (F.E.G.B.);; 2Princess Margaret Cancer Centre, University Health Network, Toronto, ON M5G 2C4, Canada; 3Toronto General Hospital, Toronto, ON M5G 2C4, Canada; 4Department of Surgery, University of Toronto, Toronto, ON M5G 2C4, Canada; 5Eliot Phillipson Clinician-Scientist Training Program, University of Toronto, Toronto, ON M5S 3H2, Canada; 6Ottawa Hospital Research Institute, Ottawa, ON K1Y 1J8, Canada; 7Department of Gastroenterology, Hepatology, Infectious Diseases and Endocrinology, Hannover Medical School, Carl-Neuberg-Straße 1, 30625 Hannover, Germany; 8St. Vincent’s University Hospital, School of Medicine, University College Dublin, Dublin D04 T6F4, Ireland; 9Department of Medical Biophysics, University of Toronto, Toronto, ON M5G 2C4, Canada; 10Institute of Medical Science, University of Toronto, Toronto, ON M5G 2C4, Canada; 11Institute for Clinical Evaluative Sciences, Toronto, ON M4N 3M5, Canada

**Keywords:** genomics, precision medicine, healthcare inequity, clinical outcomes, therapeutic access

## Abstract

Comprehensive molecular profiling can identify alterations in biliary tract cancer (BTC) potentially treatable with targeted therapies. However, the impact of whole-genome and transcriptome sequencing (WGTS) on therapeutic decision-making in a public healthcare system is unknown. Here, BTC patients prospectively received WGTS to inform clinical care at a large Canadian academic cancer center. We characterized the proportion of targetable alterations, the treatment recommendations generated by a molecular tumor board, targeted therapies received, patient outcomes, and the financing of these treatments. A total of 55 patients with BTC prospectively underwent WGTS to inform clinical care. Of those 55, 28 (51%, 95% CI 38–64%) harbored targetable alterations. Molecular tumor boards recommended consideration of targeted therapies for 43 (78% CI: 66–87%) of 55 cases. Among the 15 patients who progressed to second-line therapy and harbored targetable alterations, 8 received nine targeted therapies. No targeted therapies were funded through the public system, and most therapies were funded through compassionate access programs from companies. These results highlight the challenges and potential for inequities when implementing precision oncology in a publicly funded healthcare system.

## 1. Introduction

Biliary tract cancers (BTCs) comprise aggressive tumors that arise from the biliary tree (cholangiocarcinoma, CCA) or gallbladder (gallbladder carcinoma). Over recent decades, both incidence and mortality rates of BTCs increased globally while the five-year survival rate remains dismal, between 7–20% [1]. BTC is often asymptomatic in the early stages, resulting in about 70% of patients being diagnosed with advanced disease [2] when curative surgery is no longer an option. For these patients, systemic therapy is the primary treatment. Based on the findings from two positive phase 3 trials, TOPAZ-01 [3] and KEYNOTE-966 [4], most guidelines now recommend immune checkpoint inhibitors (ICIs) combined with chemotherapy as the standard first-line regimen [5]. However, objective response rates are only 27–29%, and median survival is around one year [3,4]. Improved treatments are urgently needed.

Targeted therapies are now standard-of-care treatment options for BTCs. Next-generation sequencing (NGS) technologies can identify targetable alterations in up to 40% [6,7,8] of BTCs. NCCN and ESMO recommend, at minimum, testing for fusions in *FGFR2* and *NTRK*, mutations in *IDH1* and *BRAF*, over-expression or amplifications of HER2/*ERBB2*, tumor-mutational burden (TMB), and microsatellite instability (MSI).

Unfortunately, barriers to accessing NGS or targeted therapies remain. In the United States, the FDA approved ivosidenib for *IDH1* mutations and pemigatinib, infigratinib, and futibatinib for *FGFR2* fusions, alongside tumor-agnostic treatments for mismatch repair deficiency, *NTRK* fusions, *ERBB2* amplifications, *BRAF* V600E mutations, and *RET* fusions. In a private healthcare system like the United States, access to NGS and therapies requires private pay or reimbursement through healthcare insurance, which can create socioeconomic discrepancies in treatments and outcomes [9].

How NGS and targeted therapies influence care in publicly funded healthcare systems is poorly understood. In theory, universal public healthcare coverage could increase access and equity to testing and treatment for BTC. However, effective implementation must respond to rapidly changing evidence and involves deciding which treatments to reimburse with a limited budget. For example, Canadian residents receive publicly funded universal healthcare insurance, and each province decides whether to fund specific cancer tests and treatments. For BTC, pemigatinib is not reimbursed by most provinces despite Health Canada approval [10]. When unavailable publicly, patients can receive targeted therapies through access programs from pharmaceutical companies, private insurance, or self-pay. Publicly funded systems in Europe and around the world face similar challenges [11].

Here, we present a prospective cohort study of patients with BTCs who received whole-genome and transcriptome sequencing (WGTS) to inform clinical care in the publicly funded healthcare system of Ontario, Canada. We characterize the recommendations generated by WGTS from a molecular tumor board, their impact on therapy, and the financing of targeted treatments.

## 2. Materials and Methods

### 2.1. Study Design and Population

The LeGresley Biliary Registry prospectively recruits all consenting adult patients (age over 18 years) diagnosed with BTC at the University Health Network from November 2021 onwards. From November 2021, all patients with advanced disease, aged 55 years or younger or with underlying inflammatory disorders (at any age) were eligible. From May 2023, all BTC patients were eligible for WGTS. Patients were identified for recruitment through screening of clinic lists. We prospectively collected demographic and clinical data from the time of enrollment. The analysis included all patients in the registry until 1 August 2024.

### 2.2. Whole-Genome and Transcriptome Sequencing

Fresh and archival tissue samples were collected for WGTS as part of the Marathon of Hope Cancer Center Network (MOHCCN; https://www.marathonofhopecancercentres.ca/; accessed on 26 January 2025). All cases underwent laser-capture microdissection (LCM). In cases where RNA failed quality metrics for whole-transcriptome sequencing, only WGS was performed. Dual DNA/RNA extraction from fresh frozen tissue and DNA for blood (buffy coat) was performed using the Qiagen Puregene Blood Kit and/or the Qiagen Allprep DNA/RNA FFPE Kit (Germantown, MD, USA) and quantitated using the Qubit 4.0 instrument (ThermoFisher Scientific, Waltham, MA, USA). Whole-genome libraries were prepared using the KAPA Hyper Prep kit (Roche, Cape Town, South Africa) with DNA extracted from fresh frozen tissue (for tumor samples) and buffy coat blood specimens (for matched normal blood samples). Then, 2 × 150 paired-end sequencing was performed using the Illumina NovaSeq6000 technology at the Ontario Institute for Cancer Research. The Illumina TruSeq Stranded Total RNA Library Prep Gold kit (San Diego, CA, USA) created whole-transcriptome libraries. Transcriptome libraries were sequenced using 2 × 100 paired reads on an Illumina NovaSeq6000 system to a target depth of 100 M PE reads (clusters) per sample.

WGTS bioinformatics were modified from those of Chan-Seng-Yue et al. [12]. Reads were mapped to the human reference genome GRCh38.p13 using BWA v.0.7.17 [13]. Duplicates were marked, and lanes were merged using Picard MarkDuplicates (v.2.21.4) (http://broadinstitute.github.io/picard/, accessed on 18 June 2024). Somatic Single Nucleotide Variants (SNVs) were called using GATK’s mutect2 (v.4.1.2) [14] and Strelka2 (v.2.9.10) [15], and calls were merged. Small insertions and deletions were called from SvaBa (v.134) [16], as well as mutect2 and strelka2, and the variants were merged. Structural variants were called using manta (v.1.6.0) [17], Delly2 (v.1.0.3) [18], and SvaBa and then merged; only variants with support from at least two callers were retained. After segmentation and relative CNV calls with HMMcopy (v.0.1.1), ploidy and sample cellularity were estimated jointly using Celluloid (v0.11.7) [19], which also calls the absolute copy number. Deletions were defined as absolute copy number less than 0.5, and amplifications as copy number greater than 4 times ploidy.

Variants were annotated with ANNOVAR (v.20170716) [20]. Tumor-mutational burden (TMB) was estimated as a U/L, where U is the mutations (i.e., SNVs and small in/dels) in the sample, and L is the length of the assessed region; we estimated both exome- and genome-wide TMBs (estimated L of 37 and 3000 Mb respectively). Mismatch repair deficiency (MMRD) was defined using immunohistochemistry. Homologous Recombination Deficiency (HRD) was called according to the hallmarks of Golan et al. 2021 [21]. Research-grade genomic reports were made for each case using djerba v1.5.1 [22]; an example djerba case report is shown in Appendix A.

Variants were annotated in OncoKB as follows: level 1: “FDA-recognized biomarker predictive of response to an FDA-approved drug in this indication”; level 2: “Standard care biomarker recommended by the NCCN or other professional guidelines predictive of response to an FDA-approved drug in this indication”; level 3A: “Compelling clinical evidence supports the biomarker as being predictive of response to a drug in this indication”; and level 3B: “Standard care or investigational biomarker predictive of response to an FDA-approved or investigational drug in another indication”. OncoKb level 4 variants (i.e., supported by “Biological Evidence”) were not annotated as targetable. The cohort was prospectively analyzed for biomarker eligibility for MTB discussion and again, retrospectively, to be compared to OncoKb recommendations (https://www.oncokb.org/therapeutic-levels, accessed on 14 March 2024). Results from OncoKB were compared quantitatively to those from MTBs.

### 2.3. Molecular Tumor Boards

The LeGresley Biliary Molecular Tumor Board (MTB) reviews all BTC patients when WGTS becomes available. The dual aims are to (1) generate consensus clinical recommendations for patients, specifically about the molecular results, and (2) inspire scientific hypotheses to guide research. The MTB currently convenes monthly and includes a team of medical oncologists, pathologists, radiologists, surgeons, bioinformaticians, genome scientists, genetic counselors, pharmacists, study nurses, and molecular biologists. Anonymized patient histories, histopathology, radiology, and genomic results are presented during the MTB meetings. Whole-genome and transcriptome sequencing, as well as additional testing such as germline NGS panels and immunohistochemistry, are reviewed to identify relevant genomic changes. Each genomic alteration is evaluated for its functional impact, oncogenic potential, and the availability of targeted therapies or relevant clinical trials. The MTB’s recommendations can include standard therapies, referrals to clinical trials, and targeted therapy with FDA- or Health Canada-approved on-label or off-label options. When a targeted therapeutic recommendation is made, priority is given to on-label options, followed by clinical trials, and then off-label options supported by the highest level of evidence. MTB considerations are documented, as outlined in Appendix A. Currently, there are no universally defined standards for the implementation, execution, and impact tracking of MTBs; however, the LeGresley Biliary MTB adheres to best practices recommended in the literature [23,24].

### 2.4. Statistical Analysis

Categorical variables were reported as numbers and percentages, and continuous variables were reported as medians and interquartile ranges (IQR). Cohort profiling analysis was performed using R software (R Foundation for Statistical Computing v4.3.1, Vienna, Austria). Binomial confidence intervals of 95% for proportions were calculated using DescTools (v0.99.54) using a two-sided Wilson test.

## 3. Results

### 3.1. Cohort Profile

A total of 55 patients had prospective WGTS to inform clinical care as part of the Registry. Of those, 53 WGTS-assessed cases passed quality metrics, while 2 cases failed the bioinformatic cellularity reporting threshold of 30%. The median time from biopsy or surgery to report was 77 days (IQR: 47–107).

Most patients were Caucasian, male, and aged 50–60 years (Table 1). Intrahepatic CCA was the most frequent cancer type, accounting for 69.1% (38/55) of cases, followed by perihilar CCA (12.7%, 7/55), gallbladder cancer (10.9%, 6/55), and distal CCA (7.3%, 4/55). At diagnosis, the median carbohydrate antigen (CA) 19-9 level was 46 U/mL (IQR: 21–282). A total of 36.4% (20/55) of patients were diagnosed with metastatic disease, while 63.6% (35/55) had localized disease and underwent surgery. Of those who had surgery, 62.9% (22/35) received adjuvant therapy, 51.4% (18/35) experienced recurrence, and 88.9% (16/18) received subsequent systemic therapy (Appendix A).

### 3.2. Targetable Alterations

The targetable alterations were diverse (Figure 1). Based on OncoKB classifications, we observed 12 different targetable genes or markers across 28 samples, including six cases (of 28; 21% [95% CI: 10–40%]) showing more than one targetable alteration. We also observed a diversity among the types of alterations (RNA fusions, amplifications and deletions, single base substitutions, and complex markers like TMB and MMRD), highlighting the utility of whole-genome and transcriptome profiling.

Overall, 24% (13/55 [95% confidence interval (CI): 14–36%]) had OncoKB level 1 targetable alterations, 29% (16/55 [95% CI: 19–42%]) had OncoKB level 1 or 2 targetable alterations, and 51% (28/55 [95% CI: 38–64%]) had OncoKB level 1–3 targetable alterations (Appendix A). The gene with the most frequent level 1 actionable mutations in our cohort was *IDH1* with hotspot mutations (n = 5, 9% [95% CI: 4–20%]). *FGFR2* fusion was the next most common (n = 4, 8% [95% CI: 3–18%]); exonic TMB-H was observed in an equal number of cases, with no overlap with *IDH1*- or *FGFR2*-mutant samples. Two cases of TMB-H also had somatic mismatch repair (MMR) deficiency (without germline hits to the MMR pathway). *ERBB2* amplification was the only level 2 alteration, which was found in the WGS data in three patients (n = 3, 5% [95% CI: 2–15%]). One patient conducted HER2 immunohistochemistry and fluorescent in situ hybridization on their tumor at another center (this was unavailable at our center at the time) and found “2+” evidence of protein overexpression and evidence of amplification (HER2/CEP17 ratio: 6.5), while the WGTS for this patient’s tumor found no alterations of *ERBB2*.

### 3.3. Molecular Tumor Board

The MTB reviewed all 55 patients and recommended considering targeted therapy for 43/55 patients (78% [95% CI: 66–87%]; Figure 2). If cases are subset to only include patients recruited after eligibility was broadened to any BTC patient (23 patients), recommendation rates remain similar (18 recommendations; 78% [95% CI 58–90%]). In 34 out of the full 55 patients in the case cohort (62% [95% CI: 49–73%]), OncoKB and the MTB recommendations agreed on the presence (or absence) of a targetable alteration (Appendix A). In 17 of the 55 cases (31% [95% CI:20–44%]), the MTB recommended marker-based clinical trials or potential off-label treatments where no OncoKb level 1–3 marker was identified (Appendix A). The most common MTB-specific recommendation was a PRMT5 inhibitor on trial for *MTAP* deletions (6/17, 35%, [95% CI: 17–59%]). Other examples included PARP inhibitors for two cases of somatic HRD without canonical HRD pathway knockout mutations (one *RAD51B* inversion and one of unknown etiology), two recommendations for MEK inhibitors (one *NF1* missense SNV and one *RASGRF2* fusion), an MDM2-inhibitor for an *MDM2* amplification, and an mTOR-inhibitor for an *STK11* missense SNV. *PTEN* deletions were observed in three cases (6% [95% CI: 2–15%]); although annotated as level 3 in OncoKB, given the absence of evidence regarding the efficacy of AKT-inhibitors in BTC, AKT-inhibitors were not consistently recommended by the MTB (*PTEN* deletion was only recommended as actionable in one of three cases). Similarly, *KRAS* p.G12D mutations (OncoKB level 2) were identified in two cases but only associated with a KRAS inhibitor recommendation from the MTB in the second case; KRAS inhibitors were too new for recommendation in the first case (MTB meeting November 2022).

### 3.4. Treatments and Outcomes

Of 43 patients with MTB recommendations for targeted therapies, 19 (44%, [95% CI: 30–59%]) had not had recurrence or progression through first-line therapy during follow-up and therefore had no indication for targeted therapy. Four patients did not survive until the MTB meeting: one died two days post-op from surgical complications, while the other three died between 2.0–9.9 weeks after biopsy and before the biomarker reports were issued. Additionally, two and three patients received supportive care alone in the first- and second-line settings, respectively. Of the remaining 15 with MTB recommendations for targeted therapies, 8 (53% [95% CI: 30–75%]) received targetable therapy in the first or second line (Figure 2).

Outcomes varied depending on whether targeted therapy was administered to the 15 eligible patients (Figure 3A). Two patients received first-line targeted treatment. One received an FGFR2 inhibitor for an *FGFR2* fusion via compassionate access in the first-line setting because of concerns around toxicity with chemotherapy. The patient was on pemigatinib for 12 months (Figure 3B). The second received durvalumab for TMB-H without MSI. Initially, the TMB-H status was unknown, and durvalumab was omitted because of a diagnosis of inflammatory bowel disease. However, after the review of the WGTS results by the MTB, durvalumab was added to gemcitabine and cisplatin in the setting of a rising CA 19-9 and mild growth on computed tomography, leading to a 27-month response.

Six patients (of 15, 40% [95% CI: 20–64%]) received targeted therapies in the second or third line. One patient received second-line pemigatinib with a response lasting for 18 months. Another patient was treated with nivolumab, given as monotherapy for recurrent melanoma that synchronously occurred with the mismatch repair deficient CCA; first-line immunotherapy was not administered because it was not accessible at the time through public funding. No benefit was observed. The third patient was given everolimus for an *STK11* mutation, without benefit. The fourth patient was given second-line FOLFOX and trastuzumab, followed by trastuzumab deruxtecan for HER2-positive IHC/FISH without *ERBB2* amplification, without response to either. The sixth patient received an experiment agent for *MTAP* deletion as part of a trial, without benefit. The final patient just started selpercatinib for *RET* overexpression at the time of data lock.

In seven cases, the MTB recommended targeted therapies after first-line therapy but the patient received chemotherapy. Targetable alterations in this group included an *FGFR2* fusion, an *ERBB2* amplification, three with OncoKB level 3B variants (two with *MDM2* amplifications and one with a *PTEN* deletion), and two with OncoKB level 4 variants (two *MTAP* deletions). Among the seven patients with targetable alterations who did not receive targeted treatment, treatment failure of second-line chemotherapy occurred within 1–13 months.

### 3.5. Funding of Therapy

The public healthcare system funded none of the nine targeted treatments across the eight patients. A total of 6/9 (67% [95% CI: 35–88%]) targeted therapies were funded compassionately through programs from pharmaceutical companies. The second-line nivolumab for MMRD CCA was funded publicly for melanoma, not CCA. The patient with HER2 overexpression received trastuzumab with FOLFOX through a compassionate program and then paid privately for trastuzumab deruxtecan in the third line.

## 4. Discussion

In this study, we performed comprehensive WGTS profiling of 55 biliary tract tumors followed by MTB review in a Canadian, publicly funded healthcare system. A total of 51% of tumors harbored targetable alterations, and 78% of tumors had targeted therapies recommended by the MTB. Among 15 patients with targetable alterations who progressed to receive second-line therapy, 53% were treated with targeted therapies; however, none were publicly funded. Among those who did not access targeted therapy, one patient had an *FGFR2* fusion, and another had an *ERBB2* amplification. Together, these results demonstrate the prospects of WGTS and MTBs to identify potentially targetable alterations, but they also highlight challenges with access to targeted therapies within publicly funded healthcare systems.

Expanding on previous studies [25], our findings reveal a high diversity of targetable alterations, including 12 genes or markers associated with precision drugs. The landscape of targets was highly diverse across genes and types of alterations. The variety of types of targetable alterations highlights the advantages of WGTS, which detects mutations, copy number alterations, fusions, overexpression, and genome-wide measurements like HRD and MSI. Furthermore, our findings demonstrate potential added value from an MTB over automated annotation. The MTB identified 20 additional potential targets over OncoKB, often for clinical trials.

While half of the profiled tumors had alterations targetable with clinically tested therapies, the expected efficacy of these therapies in biliary tract cancers varies. The efficacy of ivosidenib for targeting *IDH1* mutations and of pemigatinib or futibatinib for *FGFR2* fusions is well established [7,26,27]. Several trials have also demonstrated durable clinical benefit of therapies targeting HER2 in BTC, based on HER2+ immunochemistry [28,29,30]; whether WGTS-based evaluation of *ERBB2* translates to similar clinical outcomes remains less clear [30]. Pan-cancer trials for therapies targeting *BRAF* mutations have included BTC patients: the ROAR trial testing dabrafenib plus trametinib in *BRAF* V600E-mutated cancers found promising results (n of BTC cases = 43 [31]), while the BEAVER trial found only modest clinical activity for binimetinib and encorafenib for non-V600E tumors (BTC n = 6 [32]). Brigimadlin for *MDM2* amplifications showed encouraging preliminary efficacy in the Brightline-2 trial [33], and the KRYSTAL-1 trial found that adagrasib demonstrated promising clinical activity for targeting *KRAS* G12C [34]. The utility of alterations in *PTEN*, *PIK3CA*, *MET*, or *EGFR* has yet to be evaluated in BTCs. Considering markers with experimental associations to targeted therapies, recent data on the trial of AMG-193 for *MTAP*-deleted BTC showed encouraging antitumor activity [35], while preliminary data for PARP-inhibitors in BRCA-associated CCA suggest the use of conducting a prospective basket trial [36].

Our results demonstrate that systematic genomic profiling in a public healthcare system can frequently identify targetable alterations; however, targeted therapies were not accessible through public funding mechanisms, increasing the risk for inequities in care in Canada. Despite access to a robust trial infrastructure, enrolling patients with mutations targeted in BTC for new clinical trials was challenging; only one patient in our cohort was recruited to a clinical trial. Most targeted therapies in BTC (*IDH1*, *FGFR2*, *ERBB2*/Her2, and MSI-H) are established as standard of care, so trials were unavailable. Instead, when available, most targeted therapies were funded through compassionate-use programs provided by pharmaceutical companies. Such programs are not universally known among healthcare teams, the applications are onerous, and the pharmaceutical companies ultimately decide eligibility. Other targeted therapies were funded by private insurance or self-payment, which may place a financial burden on patients and is not an option for many.

Our results demonstrate that patients face systemic barriers to accessing targeted therapies in Canada. Similar regulatory and funding barriers likely prevent or delay access to targeted therapies for BTC in other publicly funded healthcare systems, for example, across the European Union [11]. These considerations suggest that the successful implementation of targeted therapies in publicly funded healthcare systems will require efficient pathways for companies seeking regulatory approval, such as through project Orbis [37]. Furthermore, innovative approaches to evaluating the value of therapies may be required for rare molecular subsets of rare cancers, where randomized trials are infeasible.

Because the feasibility of precision cancer medicine depends on its cost-effectiveness, several studies have tested and confirmed the cost-effectiveness of NGS in advanced solid tumors [38,39,40,41]. To evaluate the benefits and costs of these novel diagnostic tools, the OCTANE clinical trial compared participants who underwent NGS testing for advanced solid tumors with propensity-score-matched patients who did not receive large gene panel testing [38]. The analysis revealed that, while multi-gene panel testing increased healthcare costs, it was also associated with improved OS in BTC patients in Ontario, Canada. Further, NGS was shown to lower publicly funded drug costs because of clinical trial enrollment and fewer in-hospital deaths. Although the direct costs of NGS assays have decreased in recent years, the therapies themselves remain a significant cost [42]. Economic analyses of pemigatinib and ivosidenib within publicly funded healthcare systems have yielded varying results [43,44,45,46]. Collaborative efforts between policymakers, healthcare systems, and pharmaceutical companies will be necessary to ensure innovative and effective treatments are sustainably and equitably accessible across healthcare systems.

Our study has several limitations, primarily due to sample size. A second limitation is the subjective nature of the MTB. We observed some variation among recommendations, such as discrepant recommendations around *PTEN* deletions. This limitation highlights the need for increased standardization of the MTB processes [47].

In conclusion, this study showed that prospective WGTS with MTB for BTCs in a publicly funded healthcare system frequently detects potentially targetable alterations. However, accessing targeted therapies depends mainly on pharmaceutical company access programs or private funding. These results suggest that successful and equitable implementation of targeted therapy for BTC internationally will require innovative regulatory and funding mechanisms that can rapidly incorporate clinical evidence for rare molecular subsets of a rare cancer. Otherwise, the rapid advancement of precision medicine could significantly exacerbate global disparities in quality cancer care.

## Figures and Tables

**Figure 1 curroncol-32-00080-f001:**
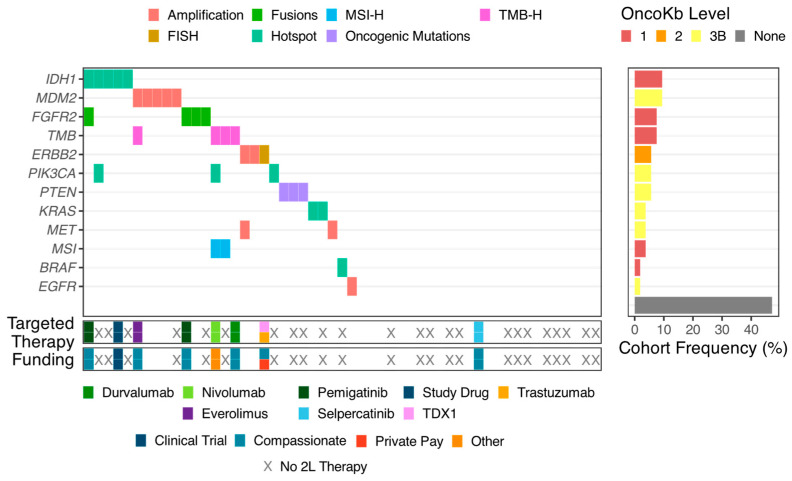
Genomics for biliary tract cancer (BTC), its actionability, and the patient’s outcome. Biomarkers that were discovered through whole-genome and transcriptome sequencing for each case (center panel, color as marker type) and their level of actionability (right panel, color as OncoKB-level). Specific targeted therapy received and funding source (bottom two panels) for each case with actionable biomarkers; cases where no targeted therapy was used because the patient did not progress to second-line therapy marked with an X.

**Figure 2 curroncol-32-00080-f002:**
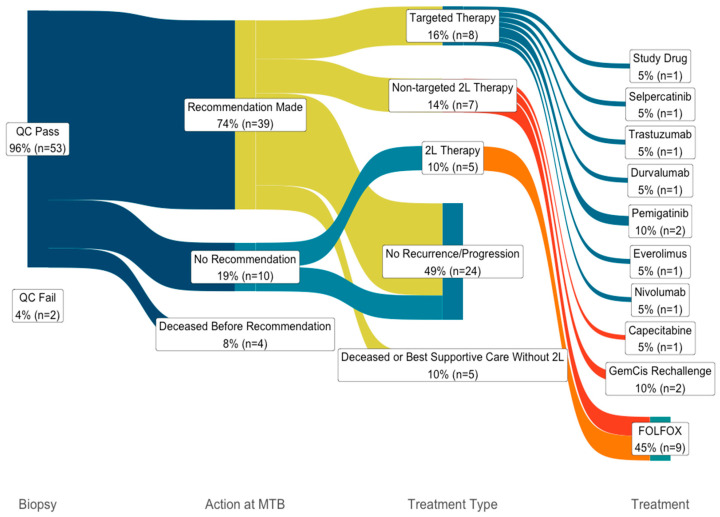
Flow of patients through four steps of the LeGresley Biliary Registry (LBR) study. Colors represent paths. Number of patients and proportion of patients in that group are labeled in respective boxes.

**Figure 3 curroncol-32-00080-f003:**
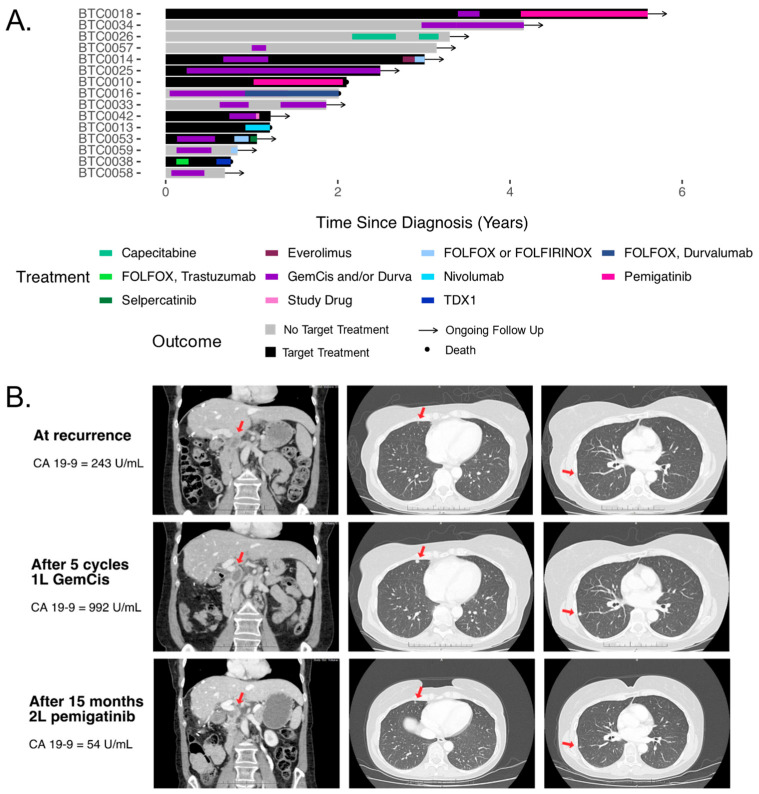
Outcomes after targeted treatments. (**A**) Swimmer’s plot showing treatments and outcomes. Patients with targetable alterations who received non-targeted second- or later-line therapies are shown in gray, while those who did receive targeted treatments are in black. Dots indicate death, while arrows indicate ongoing follow-up. GemCis: gemcitabine and cisplatin. FOLFOX: 5-fluorouracil and oxaliplatin, FOLFIRINOX: 5-fluorouracil, irinotecan, and oxaliplatin. (**B**) Serial carbohydrate antigen 19-9 (CA-19-9) and computed tomography images from BTC0010. The red arrow highlights the presence of recurrent tumor lesions.

**Table 1 curroncol-32-00080-t001:** Baseline characteristics.

	Overall, n = 55
Male sex (male, n, [%])	33 (60.0)
Age at diagnosis, years (IQR)	53 (45–69)
Body mass index, kg/m^2^ (median, IQR)	25.2 (22.9–30.1)
Ethnicity (n, [%])	
Caucasian	34 (61.8)
Asian	10 (18.2)
Black/African Canadian	5 (9.1)
Unknown	6 (10.9)
Anatomic origin of the cancer (n, [%])	
Intrahepatic	38 (69.1)
Perihilar	7 (12.7)
Distal	4 (7.3)
Gallbladder	6 (10.9)
Cancer stage, AJCC 8th (n, [%])	
I	4 (7.3)
II	16 (29.1)
IIIA	4 (7.3)
IIIB	7 (12.7)
IV	23 (41.8)
CA-19-9 at diagnosis (U/mL, IQR)	46 (21–282)
Metastasis at diagnosis (n, [%])	20 (37.7)
Surgical resection (n, %)	35 (63.6)
Adjuvant chemotherapy (n/35, [%])	
Capecitabine	17 (48.6)
Gemcitabine and cisplatin	4 (11.4)
Gemcitabine	1 (2.9)
None	13 (37.1)
Recurrence after surgery (n/35, [%])	18 (51.4)
First-line systemic therapy (n/37, [%])	
Gemcitabine, cisplatin, and durvalumab	25 (67.6)
Gemcitabine and cisplatin	6 (16.2)
Gemcitabine, cisplatin, and nab-paclitaxel	1 (2.7)
Capecitabine	2 (5.4)
FOLFIRINOX	1 (2.7)
FOLFOX	1 (2.7)
Pemigatinib	1 (2.7)
Second-line systemic therapy (n/19, [%])	
FOLFOX	11 (57.9)
Capecitabine	1 (5.3)
Pemigatinib	1 (5.3)
Everolimus	1 (5.3)
Trastuzumab	1 (5.3)
Nivolumab	1 (5.3)
AMG 193	1 (5.3)
Selpercatinib	1 (5.3)
Clinical trial	1 (5.3)

Abbreviations: IQR: interquartile range, AJCC: American Joint Committee on Cancer, FOLFIRINOX: folinic acid, fluorouracil, irinotecan, and oxaliplatin, FOLFOX: folinic acid, fluorouracil, and oxaliplatin.

## Data Availability

All WGTS fastq data files are available on EGA under DAC ID: EGAC50000000528. Analysis code is available on GitHub under https://github.com/PanCuRx-OICR/lbr-mtb (data push 26 January 2025).

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
