# Peer review of "Molecular Tumor Board-Guided Targeted Treatments for Biliary Tract Cancers in a Publicly Funded Healthcare System"

_curroncol, 2025, doi:10.3390/curroncol32020080_

Round 1
Reviewer 1 Report
Comments and Suggestions for Authors
The authors review their experience with NGS to identify target lesions in patients with BTC and the accessibility and effectiveness of target therapies.
My only suggestion to the authors is if they could expand in their conclusions what they think the effect of wgs/ngs techniques in the Canadian population would yield in terms of potential patients with targeted mutations amenable to treatment and speculative cost benefit in terms of treatment success versus current therapy. I am not asking the authors to undertake the analysis themselves as I understand such an analysis could be quite extensive, however, my ask is for the authors to speculate or discuss if such analyses have been considered and reference that in their conclusions.
Author Response
Comment 1: The authors review their experience with NGS to identify target lesions in patients with BTC and the accessibility and effectiveness of target therapies.
My only suggestion to the authors is if they could expand in their conclusions what they think the effect of wgs/ngs techniques in the Canadian population would yield in terms of potential patients with targeted mutations amenable to treatment and speculative cost benefit in terms of treatment success versus current therapy. I am not asking the authors to undertake the analysis themselves as I understand such an analysis could be quite extensive, however, my ask is for the authors to speculate or discuss if such analyses have been considered and reference that in their conclusions.
Response 1: Thank you for the recommendation. We agree that the manuscript would benefit from further evaluation of the cost-benefit of precision therapy in biliary tract cancers in Canada, and this point was also brought up by reviewer 2. We have added the following paragraph considering this question:
Because the feasibility of precision cancer medicine depends on its cost-effectiveness, several studies have tested and confirmed the cost-effectiveness of NGS in advanced solid tumors[27–30]. To evaluate the benefits and costs of these novel diagnostic tools, the OCTANE clinical trial compared participants who underwent NGS testing for advanced solid tumors with propensity-score-matched patients who did not receive large gene panel testing. The analysis revealed that, while multi-gene panel testing increased healthcare costs, it also was associated with improved OS in BTC patients in Ontario, Canada[27]. However, NGS was also associated with lower publicly-funded drug costs because of clinical trial enrolment and fewer in-hospital deaths. Although the direct costs of NGS assays have decreased in recent years, the therapies themselves remain a significant cost[31]. Economic analyses of pemigatinib and ivosidenib within publicly-funded healthcare systems have yielded varying results [32-35]. Collaborative efforts between policymakers, healthcare systems, and pharmaceutical companies will be necessary to ensure innovative and effective treatments are sustainably and equitably accessible across healthcare systems.
Reviewer 2 Report
Comments and Suggestions for Authors
This study addresses an important topic on the necessity of WGTS-based targeted therapy following first-line therapy and the role of a publicly funded healthcare system in supporting such interventions.
The process of identifying targeted treatments through Molecular Tumor Board reviews is presented as a compelling and rational approach, which is well-articulated in the manuscript.
However, there are a few areas that require further elaboration for the paper to reach its full potential:
1. Effectiveness of Targeted Therapy: The manuscript provides limited explanation regarding the therapeutic efficacy and clinical benefits of the proposed targeted therapies. Including more detailed evidence or case-based outcomes to demonstrate the effectiveness of these therapies would strengthen the argument significantly.
2. Public Benefit and Justification of Funding: The discussion on whether the allocation of public funds for these targeted therapies aligns with the broader public interest is insufficient. Expanding on how this approach could provide equitable and substantial benefits within a publicly funded healthcare system would enhance the paper’s relevance and impact.
With these improvements, I believe this manuscript could merit acceptance following minor revisions.
Author Response
Comment 1. Effectiveness of Targeted Therapy: The manuscript provides limited explanation regarding the therapeutic efficacy and clinical benefits of the proposed targeted therapies. Including more detailed evidence or case-based outcomes to demonstrate the effectiveness of these therapies would strengthen the argument significantly.
Response 1: Thank you for the recommendation. We agree the article would indeed benefit from further review of the clinical landscape of precision therapies in biliary tract cancers. We have added the following paragraph:
While half of the profiled tumors had alterations targetable with clinically tested therapies, the expected efficacy of these therapies in biliary tract cancers varies. The efficacy of ivosidenib for targeting IDH1 mutations and of pemigatinib or futibatinib for FGFR2 fusions is well established[7, 15, 16]. Several trials also demonstrate durable clinical benefit of therapies targeting HER2 in BTC, based on HER2+ immunochemistry[17–19]; whether WGTS-based evaluation of ERBB2 translates to similar clinical outcomes remains less clear[19]. Pan-cancer trials for therapies targeting BRAF mutations have included BTC patients: the ROAR trial testing dabrafenib plus trametinib in BRAF V600E-mutated cancers found promising results (BTC N = 43[20]), while the BEAVER trial found only modest clinical activity for binimetinib and encorafenib for non-V600E tumors (BTC N = 6[21]). Brigimadlin for MDM2 amplifications showed encouraging preliminary efficacy in the Brightline-2 trial[22] and the KRYSTAL-1 trial found adagrasib demonstrates promising clinical activity for targeting KRAS G12C[23]. The utility of alterations in PTEN, PIK3CA, MET or EGFR has yet to be evaluated in BTCs. Considering markers with experimental associations to targeted therapies, recent data on the trial of AMG-193 for MTAP-deleted BTC showed encouraging antitumor activity[24], while preliminary data for PARP-inhibitors in BRCA-associated CCA suggest the use of conducting a prospective basket trial[25].
Comment 2. Public Benefit and Justification of Funding: The discussion on whether the allocation of public funds for these targeted therapies aligns with the broader public interest is insufficient. Expanding on how this approach could provide equitable and substantial benefits within a publicly funded healthcare system would enhance the paper’s relevance and impact.
Response 2. Thank you for the recommendation. We agree that the manuscript would benefit from further evaluation of the cost-benefit of precision therapy in biliary tract cancers in Canada, and this point was also brought up by reviewer 1. We have added the following paragraph considering this question:
Because the feasibility of precision cancer medicine depends on its cost-effectiveness, several studies have tested and confirmed the cost-effectiveness of NGS in advanced solid tumors[27–30]. To evaluate the benefits and costs of these novel diagnostic tools, the OCTANE clinical trial compared participants who underwent NGS testing for advanced solid tumors with propensity-score-matched patients who did not receive large gene panel testing. The analysis revealed that, while multi-gene panel testing increased healthcare costs, it also was associated with improved OS in BTC patients in Ontario, Canada[27]. However, NGS was also associated with lower publicly-funded drug costs because of clinical trial enrolment and fewer in-hospital deaths. Although the direct costs of NGS assays have decreased in recent years, the therapies themselves remain a significant cost[31]. Economic analyses of pemigatinib and ivosidenib within publicly-funded healthcare systems have yielded varying results [32-35]. Collaborative efforts between policymakers, healthcare systems, and pharmaceutical companies will be necessary to ensure innovative and effective treatments are sustainably and equitably accessible across healthcare systems.